# Vaporized Hydrogen Peroxide and Ozone Gas Synergistically Reduce Prion Infectivity on Stainless Steel Wire

**DOI:** 10.3390/ijms22063268

**Published:** 2021-03-23

**Authors:** Hideyuki Hara, Junji Chida, Agriani Dini Pasiana, Keiji Uchiyama, Yutaka Kikuchi, Tomoko Naito, Yuichi Takahashi, Junji Yamamura, Hisashi Kuromatsu, Suehiro Sakaguchi

**Affiliations:** 1Division of Molecular Neurobiology, Institute for Enzyme Research (KOSOKEN), Tokushima University, Tokushima 770-8503, Japan; hara@tokushima-u.ac.jp (H.H.); jchida@tokushima-u.ac.jp (J.C.); agrianipasiana@yahoo.com (A.D.P.); ku200@tokushima-u.ac.jp (K.U.); 2Department of Nutrition, Faculty of Healthcare Sciences, Chiba Prefectural University of Health Sciences, Chiba 261-0014, Japan; yutaka.kikuchi_00@cpuhs.ac.jp; 3RD Center, Miura Corporation, Ehime 799-2651, Japan; naito_tomoko@miuraz.co.jp (T.N.); takahashi_yuichi@miuraz.co.jp (Y.T.); 4Environment & Clean Group, Strategy Development Department, Industrial Systems & General-Purpose Machinery Business Area, IHI Corporation, Tokyo 135-8710, Japan; yamamura7691@ihi-g.com (J.Y.); kuromatsu9064@ihi-g.com (H.K.)

**Keywords:** prion, prion disease, Creutzfeldt–Jakob disease, ozone gas, hydrogen peroxide gas

## Abstract

Prions are infectious agents causing prion diseases, which include Creutzfeldt–Jakob disease (CJD) in humans. Several cases have been reported to be transmitted through medical instruments that were used for preclinical CJD patients, raising public health concerns on iatrogenic transmissions of the disease. Since preclinical CJD patients are currently difficult to identify, medical instruments need to be adequately sterilized so as not to transmit the disease. In this study, we investigated the sterilizing activity of two oxidizing agents, ozone gas and vaporized hydrogen peroxide, against prions fixed on stainless steel wires using a mouse bioassay. Mice intracerebrally implanted with prion-contaminated stainless steel wires treated with ozone gas or vaporized hydrogen peroxide developed prion disease later than those implanted with control prion-contaminated stainless steel wires, indicating that ozone gas and vaporized hydrogen peroxide could reduce prion infectivity on wires. Incubation times were further elongated in mice implanted with prion-contaminated stainless steel wires treated with ozone gas-mixed vaporized hydrogen peroxide, indicating that ozone gas mixed with vaporized hydrogen peroxide reduces prions on these wires more potently than ozone gas or vaporized hydrogen peroxide. These results suggest that ozone gas mixed with vaporized hydrogen peroxide might be more useful for prion sterilization than ozone gas or vaporized hydrogen peroxide alone.

## 1. Introduction

Prion diseases are a group of fatal neurodegenerative disorders in humans and animals, caused by accumulation of infectious protein aggregates, or prions, in the brain [1,2]. They manifest as sporadic, genetic, and acquired diseases in humans [3,4]. Sporadic Creutzfeldt–Jakob disease (sCJD) is most common, accounting for about 85% of human prion disease cases [3,4,5]. Genetic prion diseases, which include familial CJD, fatal familial insomnia, and Gerstmann–Sträussler–Scheinker syndrome, are linked to pathogenic mutations in the prion protein gene *Prnp*, and account for 10–15% of human cases [3,4,6]. The remaining cases are those of acquired prion diseases, including kuru, variant CJD (vCJD), and iatrogenic CJD (iCJD). Kuru is a disease that prevails among the Fore people in Papua New Guinea through ritual mortuary rites [4,7]. vCJD is transmitted from bovine spongiform encephalopathy (BSE), particularly the classic type of BSE, through consumption of BSE-contaminated beef [4,8]. Fortunately, BSE cases have been dramatically reduced due to the meat and bone meal ban in ruminant feed and therefore the transmission risk of BSE to humans has become very low at present [9]. However, iatrogenic transmission of CJD still remains as potential public health concern. People have been reported to develop iCJD after receiving corneal or dura mater grafts, growth hormone, gonadotropin, and red blood cells that were contaminated with prions [10]. Medical instruments that were used in preclinical or clinical sCJD patients have been also shown to transmit the disease. Two patients were reported to die of iCJD after stereo-electroencephalographical examinations using the electrodes that were previously used for other sCJD patients [10,11]. Neurosurgical instruments that were used for preclinical sCJD patients have been also reported to transmit the disease [10,11,12]. There is also an epidemiological report suggesting that surgeries for peripheral vessels, digestive system, spleen, and female genital organs might be associated with iCJD risk [13]. Adequate prion decontamination procedures are therefore required for medical instruments, particularly surgical instruments, so as not to potentially transmit the disease.

Prions consist of the misfolded, amyloidogenic isoform of prion protein, designated PrP^Sc^, which is produced through conformational conversion of the normal cellular counterpart, PrP^C^, a membrane glycoprotein expressed most abundantly in the brain, particularly by neurons [1,2]. Consistent with prions being made of protein alone, protein denaturants including urea, guanidine-hydrochloride, sodium dodecyl sulfate, and sodium hydroxide have been shown to effectively reduce prion infectivity whereas nucleic acid-damaging procedures, which are capable of inactivating conventional pathogens such as bacteria and viruses, do not [14,15,16,17]. High-temperature autoclaving has been also shown to reduce prion infectivity [14,18]. However, medical instruments or devices are so complicated and delicate that they may be susceptible to the damage that the chemicals or autoclaving may cause. Effective reduction of prion infectivity has been also reported with gas sterilizations, including hydrogen peroxide gas sterilization [19,20]. Mixing of ozone solution in prion-infected brain homogenates was also shown to reduce prion infectivity in the homogenates [21]. These facts motivated us to explore if hydrogen peroxide gas could synergistically function with ozone gas for inactivation of prion infectivity.

In this study, to address the question, we exposed RML scrapie prions fixed on the surface of stainless steel wires to ozone gas alone, vaporized hydrogen peroxide alone, and ozone gas mixed with vaporized hydrogen peroxide, and implanted them into the brains of mice. Mice implanted with the wires exposed to ozone gas mixed with vaporized hydrogen peroxide developed disease significantly later than those implanted with the ozone gas- or the vaporized hydrogen peroxide only-exposed wires, indicating that ozone gas mixed with vaporized hydrogen peroxide could be more effective in reducing RML prion infectivity on stainless steel wires than ozone gas alone or vaporized hydrogen peroxide alone. These results suggest that ozone gas and vaporized hydrogen peroxide could synergistically reduce prion infectivity on stainless steel wires.

## 2. Results

### 2.1. Ozone Gas and Vaporized Hydrogen Peroxide Synergistically Reduce Prion Infectivity on Stainless Steel Wires

To investigate if ozone gas and vaporized hydrogen peroxide could synergistically inactivate prion infectivity, we exposed ozone gas, vaporized hydrogen peroxide, and ozone gas mixed with vaporized hydrogen peroxide to RML scrapie prions fixed on the surface of stainless steel wires in the endotoxin (ET) sterilization mode of the gas sterilizer (Figure 1A). The treated wires were then implanted into the brains of ICR mice. We also similarly implanted gas-unexposed, prion-contaminated stainless steel wires into ICR mice as a control. Mice implanted with the control wires developed disease at 176 ± 8 days post-implantation (dpi, 10 diseased/10 implanted mice) (Figure 1B). However, compared to those in the control mice, incubation times were prolonged in mice implanted with ozone gas-exposed, prion-contaminated wires (190 ± 16 dpi, 10 diseased/10 implanted mice, *p* < 0.0001), further elongated in mice implanted with vaporized hydrogen peroxide-exposed, prion-contaminated wires (>236 ± 16 dpi, 9 diseased/10 implanted mice, *p* < 0.0001), and greatly extended in mice implanted with prion-contaminated wires exposed to ozone gas-mixed vaporized hydrogen peroxide (>315 ± 68 dpi, 4 diseased/10 implanted mice, *p* < 0.0001) (Figure 1B). These results indicate that, while ozone gas and vaporized hydrogen peroxide alone could effectively reduce prion infectivity, they could synergistically reduce prion infectivity so strongly that more than half of the mice implanted with prion-fixed wires exposed to ozone gas mixed with vaporized hydrogen peroxide remain disease-free.

We then investigated accumulation of PrP^Sc^ in the brains of these mice by Western blotting with 6D11 anti-PrP antibody, which recognizes residues 93–109 of mouse PrP. The PK (proteinase K)-resistant PrP fragments, or PrP^res^, with similar glycosylation banding patterns, were observed at similar levels in the brains of all the diseased mice examined (Figure 2). In contrast, no PrP^res^ was observed in the brains of non-diseased mice sacrificed at 365 dpi, except for mild accumulation of PrP^res^ detected in one mouse implanted with ozone gas mixed with vaporized hydrogen peroxide-exposed, prion-fixed wires (Figure 2). We also examined prion disease-specific vacuolation in the brains of diseased mice. Many vacuoles were observed in various brain regions, particularly in the cerebral cortex, hippocampus, thalamus, and cerebellum of diseased mice, but not in prion-uninfected control brains (Figure 3A). Immunohistochemical staining also showed PrP^Sc^ accumulation throughout the brain sections from diseased mice, but not from prion-uninfected control mice (Figure 3B). These results suggest that, while ozone gas, vaporized hydrogen peroxide, and ozone gas-mixed vaporized hydrogen peroxide could reduce prion infectivity, they could not affect prion pathogenic properties such as those relevant to brain accumulation levels of PrP^Sc^ and vacuolating brain regions.

### 2.2. Prion-Inactivating Activity of Ozone Gas Mixed with Vaporized Hydrogen Peroxide Is Correlated to Exposure Time but Not to Vaporized Hydrogen Peroxide Concentration

To investigate the effects of the exposure time of ozone gas mixed with vaporized hydrogen peroxide on its inactivating activity of prion infectivity, we exposed RML prion-contaminated stainless steel wires to ozone gas-mixed vaporized hydrogen peroxide in two different time windows using the short and the standard sterilization modes (Figure 4A) and implanted them into the brains of ICR mice. Mice implanted with the standard mode-treated wires developed disease with significantly longer incubation times than those implanted with the short mode-treated wires (>309 ± 62 dpi (6 diseased/11 implanted mice) vs. >261 ± 68 dpi (10 diseased/11 implanted mice), *p* = 0.0363) (Figure 4B), indicating that longer exposure time could increase the prion-inactivating activity of ozone gas mixed with vaporized hydrogen peroxide.

We thus assumed that further longer exposure with an increasing concentration of hydrogen peroxide could enhance the prion-inactivating activity of ozone gas mixed with vaporized hydrogen peroxide. To investigate this, we exposed prion-contaminated stainless steel wires to ozone gas mixed with vaporized hydrogen peroxide in the long sterilization mode and implanted them into the brains of ICR mice. In the long mode, 45% hydrogen peroxide was used for the second injection of hydrogen peroxide and vaporized for about 13 min, instead of 3% hydrogen peroxide that was vaporized for about 9 min in the standard mode and 4 min in the short mode (Figure 4A). Paradoxically, incubation times of mice implanted with the long mode-treated wires (225 ± 36 dpi, 11 diseased/11 implanted mice) were shortened to those of mice implanted with the short mode-treated wires (*p* = 0.1206) (Figure 4B). These results indicate that long exposure to high concentration of vaporized hydrogen peroxide could diminish the synergistic effects of ozone gas-mixed vaporized hydrogen peroxide on reduction of prion infectivity. However, it remains unknown whether the reduced sterilizing activity of ozone gas-mixed vaporized hydrogen peroxide in the long mode against prions is due to the effects of long exposure, the high concentration of vaporized hydrogen peroxide, or both.

### 2.3. Ozone Gas Mixed with Vaporized Hydrogen Peroxide Reduces Prion Infectivity in a Highly Sensitive Mouse Model

We also confirmed the prion-inactivating activity of ozone gas mixed with vaporized hydrogen peroxide using a highly sensitive mouse model. Transgenic mice, designated Tg(MoPrP)/*Prnp*^0/0^ mice, are highly sensitive to prion infection due to about 4-times higher expression of mouse PrP^C^ in their brains on the *Prnp*^0/0^ genetic background than in WT mice [22]. Since the ET mode- and the standard mode-treated wires reduced prion infectivity to similar levels in ICR mice, we exposed ozone gas mixed with vaporized hydrogen peroxide to RML prions fixed on stainless steel wires in the ET and the standard modes and implanted them into the brains of Tg(MoPrP)/*Prnp*^0/0^ mice. All Tg(MoPrP)/*Prnp*^0/0^ mice implanted with the gas-unexposed, prion-contaminated wires developed disease at 96 ± 7 dpi (Table 1). However, after implantation of the ozone gas mixed with vaporized hydrogen peroxide-exposed, prion-contaminated wires in the ET or the standard mode, only 2 out of 7 Tg(MoPrP)/Prnp^0/0^ mice succumbed to disease by 326 dpi (Table 1). These results further confirm that ozone gas mixed with vaporized hydrogen peroxide in the ET mode or the standard mode could similarly reduce RML prions on stainless steel wires.

## 3. Discussion

In the present study, we compared ozone gas, vaporized hydrogen peroxide, and ozone gas mixed with vaporized hydrogen peroxide for their activities to reduce prion infectivity, by exposing RML prions contaminated on stainless steel wires to each gas and then implanting the treated stainless steel wires into the brains of mice. Consistent with the results previously reported by others showing that ozone gas and vaporized hydrogen peroxide could reduce prion infectivity [19,20,21,23,24], we found that mice implanted with RML prion-contaminated wires exposed to ozone gas or vaporized hydrogen peroxide developed the disease significantly later than those implanted with gas-unexposed, RML prion-contaminated wires. However, incubation times of mice implanted with RML prion-contaminated wires exposed to ozone gas mixed with vaporized hydrogen peroxide were much longer than those of mice implanted with RML prion-contaminated wires exposed to ozone gas alone or vaporized hydrogen peroxide alone. These results indicate that ozone gas mixed with vaporized hydrogen peroxide may be more powerful to reduce prion infectivity than ozone gas alone or vaporized hydrogen peroxide alone, suggesting that ozone gas and vaporized hydrogen peroxide could synergistically function to reduce prion infectivity.

Prion infectivity is believed to be enciphered in the protein conformation of PrP^Sc^. Upon prion infection, PrP^Sc^ interacts with PrP^C^ and induces conformational changes in the interacting PrP^C^ to adopt the conformation of PrP^Sc^, eventually causing prion disease [25,26]. Ozone and hydrogen peroxide are powerful oxidants, oxidizing susceptible amino acids such as cysteine, methionine, tryptophan, tyrosine, histidine, and phenylalanine and thereby causing the structural changes of target proteins [27,28]. Thus, ozone gas and vaporized hydrogen peroxide oxidize PrP^Sc^, modulating its protein conformation and leading to destruction of the activity to convert PrP^C^ into PrP^Sc^, or prion infectivity. Ozone gas mixed with vaporized hydrogen peroxide may oxidize PrP^Sc^ more robustly than ozone gas alone or vaporized hydrogen peroxide alone, thereby reducing prion infectivity more strongly than ozone gas alone or vaporized hydrogen peroxide alone.

Mice implanted with the standard mode-treated, RML prion-contaminated stainless steel wires developed disease with significantly longer incubation times than those implanted with short mode-treated, RML prion-contaminated wires, indicating that the standard mode is more effective in reducing prion infectivity than the short mode. Since the exposure time of ozone gas mixed with vaporized hydrogen peroxide in the standard mode is longer than that in the short mode, these results indicate that exposure time is an important factor for ozone gas mixed with vaporized hydrogen peroxide to effectively reduce prion infectivity. However, mice implanted with the long mode-treated wires developed disease with shortened incubation times, which were similar to those of mice implanted with the vaporized hydrogen peroxide alone-treated wires. In the long mode, RML prion-fixed wires were exposed to an increasing concentration of hydrogen peroxide in a long exposure period, suggesting that higher concentration of vaporized hydrogen peroxide in a long period might reduce the synergistic oxidative effects of ozone gas mixed with vaporized hydrogen peroxide on reduction of prion infectivity. It has been reported that high concentration of hydrogen peroxide could act as a scavenger for hydroxyl radicals generated during decomposition processes of ozone in water phase [29]. It might be thus possible that high concentration of vaporized hydrogen peroxide might also scavenge ozone gas-derived hydroxyl radicals in vapor phase. However, this remains to be clarified. We also found that incubation times of mice implanted with the standard mode-treated wires were not significantly elongated, compared to those of mice implanted with the ET mode-treated wires, although total exposure time is longer in the ET mode than in the standard mode. It is thus possible that the exposure time used in the standard mode might be long enough for reduction of prion infectivity by ozone gas-mixed vaporized hydrogen peroxide.

In short, we showed that ozone gas mixed with vaporized hydrogen peroxide could reduce prion infectivity fixed on stainless steel wires more strongly than ozone gas alone or vaporized hydrogen peroxide alone, suggesting that ozone gas and vaporized hydrogen peroxide could synergistically reduce prion infectivity on stainless steel wires. However, further studies are required for ozone gas mixed with vaporized hydrogen peroxide to be used in practical use as a prion decontamination tool for medical instruments or devices.

## 4. Materials and Methods

### 4.1. Ethics Statement

Animals were cared for in accordance with The Guiding Principle for Animal Care and Experimentation of Tokushima University and with Japanese Law for Animal Welfare and Care. Mice were housed under specific pathogen-free conditions in cages of 5–6 animals with water and food ad libitum. Cages were provided with a standard softwood bedding. Mice were kept on a standard 12:12 light:dark cycle. Every effort was made to reduce distress and the number of animals used according to the ARRIVE guidelines.

### 4.2. Antibodies

The antibodies used in this study are: 6D11 mouse anti-PrP Ab (SIG-399810; BioLegend, San Diego, CA, USA), mouse anti-ß-actin Ab (M177-3; Medical & Biological Laboratories Co., Ltd., Nagoya, Japan), anti-mouse IgG horseradish peroxidase (HRP)-linked Ab (NA931; GE Healthcare, Little Chalfont, UK).

### 4.3. Animals

Crl:CD1(ICR) mice were purchased from Charles River Laboratories Japan (Kanagawa, Japan). Tg(MoPrP)/*Prnp*^0/0^ mice were obtained elsewhere by intercross between the backcrossed *Prnp*^0/0^ mice and Tg(MoPrP) mice with a FVB background [22].

### 4.4. Fixation of RML Prions onto Stainless Steel Wires

Stainless wires (751267; SUS 304; diameter 0.2 mm; The Nilaco Corporation, Tokyo, Japan) were cut to 5 mm in length. Batches of 10 wires were sterilized in a 1.5 mL microtube containing 1 mL of 70% (vol/vol) ethanol in deionized water at room temperature (RT) overnight. After air drying, the stainless steel wires were incubated in a microtube containing 400 µL of 10% (*wt*/*v*) RML-infected brain homogenates at 4 °C overnight. Brain homogenates were prepared as described previously [30]. The wires were then washed three times in a microtube containing 1 mL of phosphate-buffered saline (PBS) (11482-15; Nakalai Tesque, Osaka, Japan) and air-dried in Petri dish at RT overnight.

### 4.5. Inactivation of RML Prions on Stainless Steel Wires

Ozone-mixed hydrogen peroxide gas sterilizer (Miura Co., Ehime, Japan) was used to inactivate RML prions fixed onto stainless steel wires in four different sterilization modes, the ET, the standard, the short, and the long modes, according to the manuals. The sterilizing chamber (W40.2 cm × H36.7 cm × D68.2 cm) of the sterilizer was injected with about 0.1 g of ozone gas (25,000 ppm) at the pre-treatment phase. Then, 2.0 mL of 45% hydrogen peroxide solution (pH 1.8–2.6; Mitsubishi gas chemical company, Inc., Tokyo, Japan) was injected and followed by injection with about 0.1 g of ozone gas (25,000 ppm) and 5.0 mL of 3% hydrogen peroxide solution (Mitsubishi gas chemical company, Inc.) in the ET, the standard, and the short modes or 45% hydrogen peroxide solution (Mitsubishi gas chemical company, Inc.) in the long mode at 50 °C. The injected hydrogen peroxide solution was vaporized in the chamber. Ozone gas used in this study was generated from oxygen in the air using an ozonizer (IHI Agri-Tech Co., Hokkaido, Japan) and concentrated using a Pressure Swing Adsorption apparatus (Sanyo Electronic Industries Co., Ltd., Okayama, Japan).

### 4.6. Intracerebral Inoculation with Prion-Infected Brain Homogenates 

For this step, 1% RML prion-infected brain homogenates in PBS were prepared using the brains from RML prion-infected, terminally ill mice and serially diluted with PBS. These homogenates were intracerebrally inoculated into 5- to 6-week-old ICR or Tg(MoPrP)/*Prnp*^0/0^ mice (20 µL/mouse) using a 30-gauge injection needle (HS-2739A; Dentronics Co., Ltd., Tokyo, Japan). Mice were diagnosed as sick when they developed more than five of the following features: emaciation, decreased locomotion, ruffled body hair, ataxic gait, kyphosis, priapism, upright tail, crossing leg, hind leg paresis, and foreleg paresis.

### 4.7. Intracerebral Insertion of Prion-Contaminated Stainless Steel Wires

The wire was inserted intracerebrally into 5- to 6-week-old mice (1 wire/mouse), using a 25-gauge injection needle (NN-2516R; Terumo Corporation, Tokyo, Japan).

### 4.8. Western Blotting

Western blotting was performed as described previously [30]. Samples were treated with or without PK (20 µg PK/mg proteins; Wako Pure Chemical Industries, Osaka, Japan) at 37 °C for 30 min. Total proteins were run on a SDS-polyacrylamide gel and transferred to an Immobilon-P PVDF membrane (IPVH00010, Millipore, Billerica, MA, USA). After blocking with 1% non-fat dry milk in TBST (10 mM Tris-HCl, pH7.4, containing 0.05% Tween-20, and 150 mM NaCl) at RT for 1 h, the membranes were incubated with 6D11 anti-PrP antibody (BioLegend) at 4 °C overnight in TBST containing 0.5% non-fat dry milk, washed 3 times with TBST at RT for 5 min, and incubated with anti-mouse IgG HRP-linked Ab (GE Healthcare) at RT for 2 h in TBST containing 0.5% non-fat dry milk. After washing 3 times with TBST at RT for 5 min, signals were visualized using Immobilon Western Chemiluminescent HRP substrate (WBKLS0500, Millipore) and detected by LAS-4000 mini chemiluminescence imaging system (Fuji Film, Tokyo, Japan).

### 4.9. Hematoxylin-Eosin Staining

Paraffin-embedded samples were sectioned at 5 µm. The sectioned samples were deparaffinized, rehydrated, and stained with Mayer’s hematoxylin solution (131-09665; Wako Pure Chemical Industries) at RT for 5 min and 1% eosin Y solution (051-06515; Wako Pure Chemical Industries) at RT for 2 min. After washing, the samples were mounted with Softmount (192-16301; Wako Pure Chemical Industries). Images of sample were visualized using BZ-810 (Keyence, Osaka, Japan) and analyzed with BZ-800 analyzer software (Keyence).

### 4.10. Immunohistochemistry

Then, 5 µm sliced paraffin-embedded brain samples were deparaffinized, rehydrated, and autoclaved in 1 mM HCl at 121 °C for 5 min. The samples were then washed with phosphate-buffered saline (PBS), digested with 50 µg/mL PK (Wako Pure Chemical Industries) in PBS at 37 °C for 30 min, treated with 3 M guanidine thiocyanate at RT for 10 min, and then washed with PBS again. After blocking with 5% fetal bovine serum in PBS at RT for 1 h, the samples were incubated with 6D11 anti-PrP Ab (BioLegend) at RT for 2 h, washed with PBS, and then treated with ImmPRESS REAGENT Anti-Mouse IgG (MP-7402, Vector Laboratories, Burlingame, CA, USA) at RT for 1 h. The samples were washed with PBS, and incubated with ImmPACT DAB Peroxidase Substrate (SK-4105, Vector Laboratories) for 180 s for staining. Images were visualized using BZ-810 (Keyence) and analyzed with BZ-800 analyzer software (Keyence).

### 4.11. Statistical Analysis

Incubation times were analyzed using the log rank (Mantel–Cox) test.

## Figures and Tables

**Figure 1 ijms-22-03268-f001:**
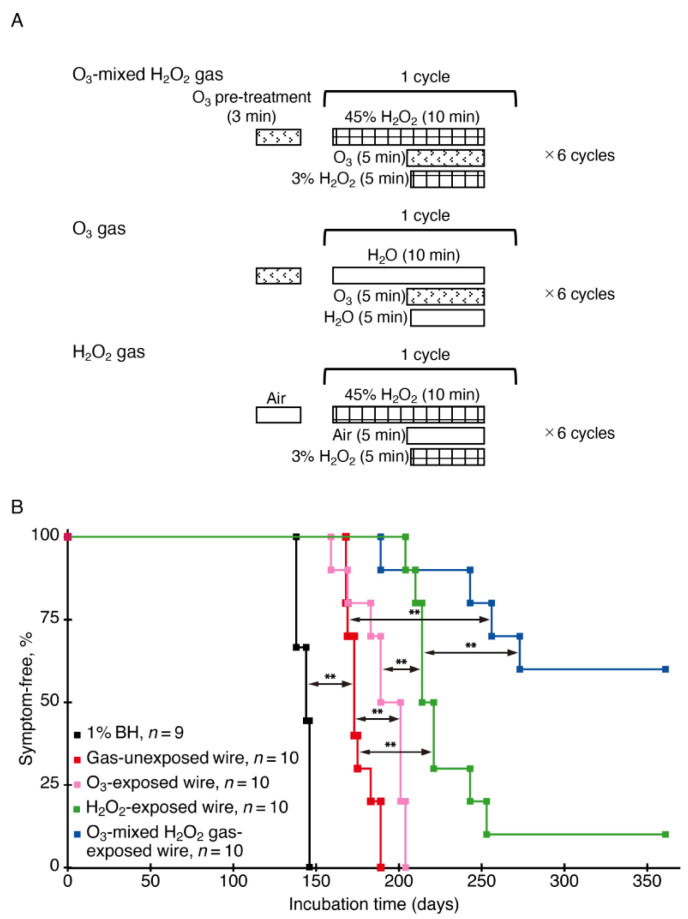
Ozone gas and vaporized hydrogen peroxide synergistically reduce prion infectivity of stainless steel wires. (**A**) Schematic representations of the sterilization protocols for ozone gas mixed with vaporized hydrogen peroxide, ozone gas alone, and vaporized hydrogen peroxide alone in the ET (endotoxin) mode. For sterilization with ozone gas mixed with vaporized hydrogen peroxide, RML prion-contaminated stainless steel wires were pre-treated by injection of 25,000 ppm ozone gas for 3 min and followed by 6 cycles of a 10 min sterilization process, which comprises injection of 45% hydrogen peroxide followed by injection of 25,000 ppm ozone gas and 3% hydrogen peroxide 5 min later. For the treatment with ozone gas alone or vaporized hydrogen peroxide alone, water was used instead of hydrogen peroxide or ozone gas, respectively. The sterilization process of each mode is terminated by injection of air. (**B**) The percentage of symptom-free mice after intracerebral inoculation with 1% brain homogenate (1% BH) from RML-infected, diseased mice and intracerebral implantation with gas-unexposed, ozone gas-exposed, vaporized hydrogen peroxide-exposed, and ozone gas mixed with vaporized hydrogen peroxide-exposed, RML prion-contaminated wires. O_3_, ozone; H_2_O_2_, hydrogen peroxide; ** *p* < 0.01.

**Figure 2 ijms-22-03268-f002:**
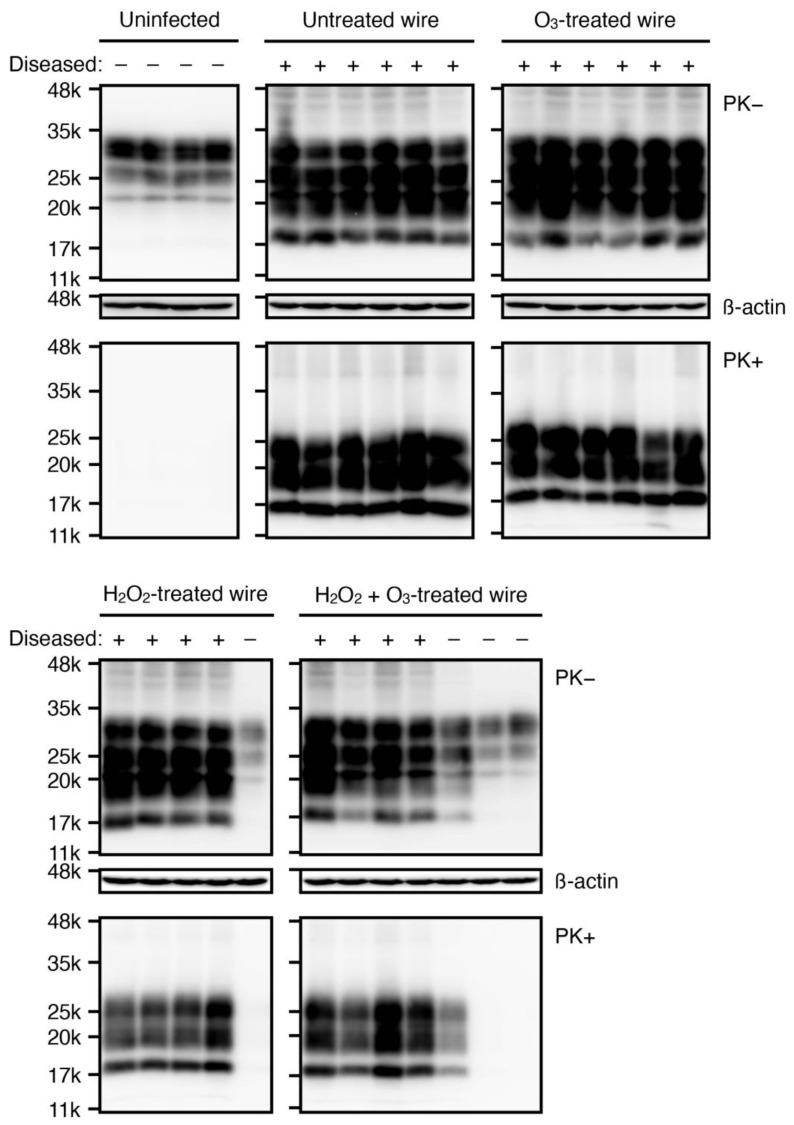
Western blotting for PrP^Sc^ in the brains of mice implanted with various gas-treated, RML prion-contaminated stainless steel wires. Brain homogenates from mice uninfected with RML prions (*n* = 4), implanted with gas-untreated, RML prion-contaminated stainless steel wires (*n* = 6), ozone gas-treated, RML prion-contaminated stainless steel wires (*n* = 6), vaporized hydrogen peroxide-treated, RML prion-contaminated wires (*n* = 5), and ozone gas mixed with vaporized hydrogen peroxide-treated, RML prion-contaminated wires (*n* = 7) were treated with or without proteinase K (PK) and subjected to Western blotting with 6D11 anti-PrP antibody. β-actin is an internal control for Western blotting. O_3_, ozone; H_2_O_2_, hydrogen peroxide.

**Figure 3 ijms-22-03268-f003:**
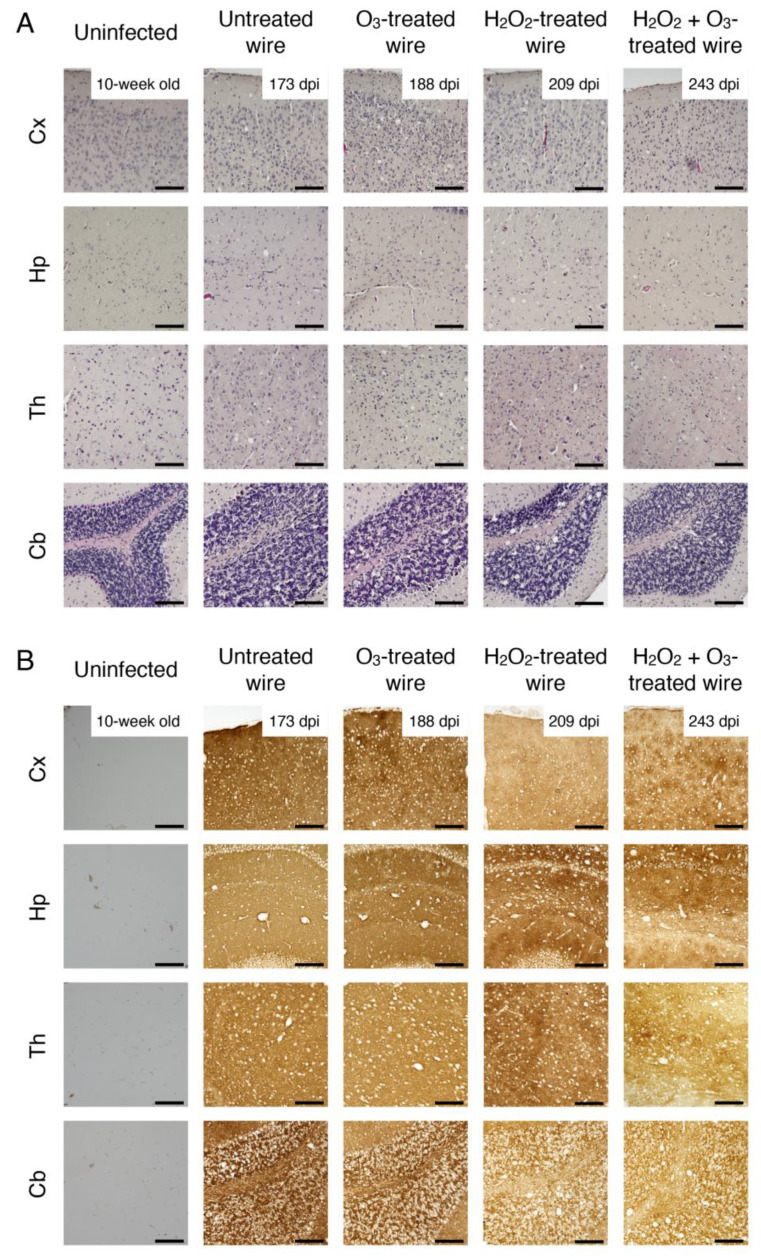
Brain pathologies in mice implanted with various gas-treated, RML prion-contaminated stainless steel wires. Representative pictures of HE-stained (**A**) and PrP^Sc^-stained (**B**) brain sections from mice uninfected with RML prions (*n* = 4, 10-week old), implanted with gas-untreated, RML prion-contaminated stainless steel wires (*n* = 4), ozone gas-treated, RML prion-contaminated stainless steel wires (*n* = 4), vaporized hydrogen peroxide-treated, RML prion-contaminated wires (*n* = 4), and ozone gas mixed with vaporized hydrogen peroxide-treated, RML prion-contaminated wires (*n* = 3) are shown. O_3_, ozone; H_2_O_2_, hydrogen peroxide. Cx, cerebral cortex; Hp, hippocampus; Th, thalamus; Cb, cerebellum. Scale bar, 100 μm.

**Figure 4 ijms-22-03268-f004:**
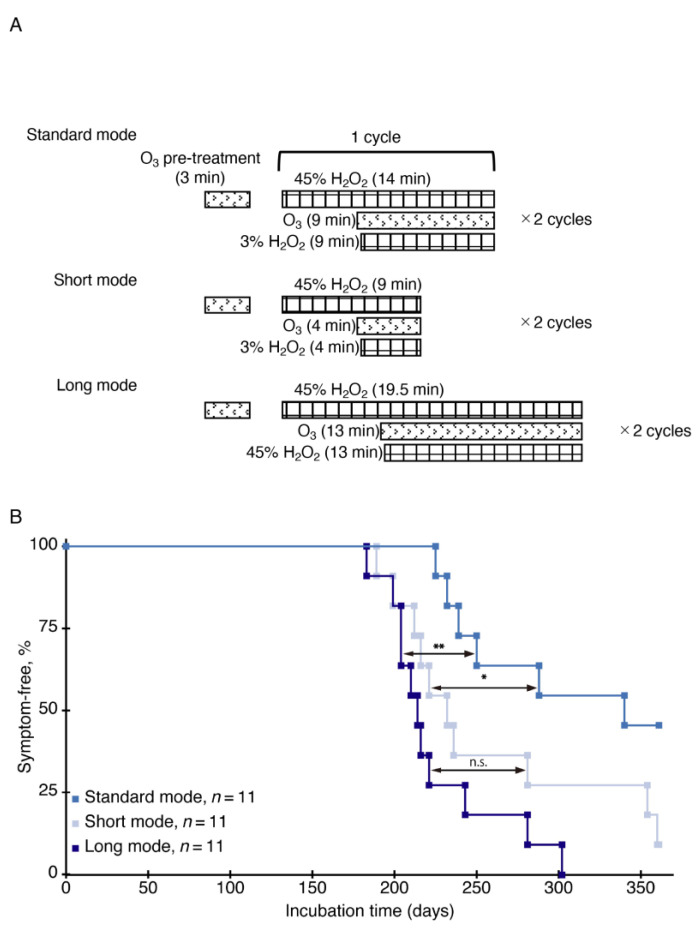
Sterilization of RML prions fixed on stainless steel wires in different sterilization modes. (**A**) Schematic representations of the sterilization protocols of ozone gas mixed with vaporized hydrogen peroxide in the standard, the short, and the long modes. RML prions-fixed wires were pre-treated by injection of 25,000 ppm ozone gas for 3 min and followed by 2 cycles of a 14-min sterilization process comprising injection of 45% hydrogen peroxide followed by injection of 25,000 ppm ozone gas and 3% hydrogen peroxide 5 min later in the standard mode, a 7-min sterilization process comprising injection of 45% hydrogen peroxide followed by injection of 25,000 ppm ozone gas and 3% hydrogen peroxide 5 min later in the short mode, and a 19.5-min sterilization process comprising injection of 45% hydrogen peroxide followed by injection of 25,000 ppm ozone gas and 45% hydrogen peroxide 6.5 min later in the long mode. The sterilization process of each mode was terminated by injection of air. (**B**) The percentage of symptom-free mice after intracerebral implantation with RML prion-contaminated stainless steel wires exposed to ozone gas mixed with vaporized hydrogen peroxide-exposed in the standard, the short, and the long modes. O_3_, ozone; H_2_O_2_, hydrogen peroxide; * *p* < 0.05; ** *p* < 0.01; n.s., not significant.

**Table 1 ijms-22-03268-t001:** Incubation times in Tg(MoPrP)/*Prnp*^0/0^ mice intracerebrally implanted with RML prion-contaminated stainless steel wires exposed to ozone gas mixed with vaporized hydrogen peroxide in the ET or standard mode.

Gas Exposure Mode	Diseased Mice /Total Mice	Incubation Times ^1^ (Days)
Unexposed	8/8	96 ± 7
Standard mode	2/7	>326 (106, 106)
ET mode	2/7	>326 (113, 116)

^1^ Implanted mice were observed by 326 dpi. Two of the mice implanted with the standard- and the ET mode-treated, prion-contaminated stainless steel developed disease at indicated dpi in parenthesis.

## Data Availability

The authors confirm that all data underlying the findings are fully available without restriction. All data are included within the manuscript.

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
