# Peer review of "Vaporized Hydrogen Peroxide and Ozone Gas Synergistically Reduce Prion Infectivity on Stainless Steel Wire"

_ijms, 2021, doi:10.3390/ijms22063268_

Round 1
Reviewer 1 Report
This is a well-designed study with importance in the field. Prion decontamination is a great concern that can be improved with the results obtained in this work.
In figure 3, HE-stained brain sections from different mice are shown. The PrPres status and survival time of the mice used in this figure is not indicated. These data would help with the interpretation of the results shown.
Reviewer 2 Report
In this manuscript Dr Hara and collaborators show a method to reduce the presence of PrPSc in stainless steel material using a mix of hydrogen peroxide and ozone. As the authors clearly show, the treatment of wires exposed to PrPSc (RML strain) reduce the presence of infectious prions compared to non-treated wires. To demonstrate it, the authors implant the wires in the CNS of WT and PrP overexpressing mice and compare the days post inoculation until the animals shows some signs of prionic disease.
The results are clearly explained and the reduction of PrPSc in the wires is demonstrated. However, before publication some minor English spell check should be done, as some phrases are not totally understandable.
My congratulations to the authors for this manuscript.
Reviewer 3 Report
The manuscript describes an interesting piece of work about the use of ozone and hydrogen peroxide, either alone or in combination, for the vapor-phase sterilization of prion-contaminated stainless steel wires.
The results cover a relevant field of application and cover a topic that is less investigated than the one of viral or bacterial sterilization.
The text is clearly written, easy to understand and rich in bibliographic references.
The contribution could therefore be positively considered for publication. However, some minor parts deserve attention and potential amelioration. In particular:
1) Throughout the text, the term “gas” and “vaporized” are used in the same way when referring to both hydrogen peroxide and ozone. However, it is likely that, under the tested conditions, hydrogen peroxide is used as a "vapor", starting from an aqueous solution of it, rather than as a "gas". Some more details should be added in Section 4.5 about the method according to which hydrogen peroxide and ozone are inserted into the gas sterilizer. Is hydrogen peroxide added as a concentrated (45%) or diluted (3%) aqueous solutions? Is ozone generated in situ from molecular oxygen? Does the process imply any change in temperature? or is it performed at room temperature?
2) line 165. During the long sterilization mode, why was 45% hydrogen peroxide used for the second injection of hydrogen peroxide, instead of 3%? Actually, this change in concentration gives rise to a double change of variables at the same time and this lead to some confusion in the interpretation of the results.
3) Is there any control of the pH of the inlet hydrogen peroxide solution? Actually, very recent observations showed that the presence of pH-modifiers in aqueous solutions of hydrogen peroxide (especially acidic additives) can dramatically increase the disinfecting capability of this active agent (see, for instance, doi: 10.1021/acs.chas.0c00095 or 10.1016/j.jhin.2020.06.001, not cited here).
4) lines 210-215. Some very long nouns are unclear and should be rephrased for the sake of clarity. For instance: “RML prion-contaminated wires exposed to ozone gas or to hydrogen peroxide gas” instead of “ozone gas- or hydrogen peroxide gas-exposed, RML prion-contaminated wires”.
5) Line 244. It is stated that high concentrations of hydrogen peroxide could act as a scavenger for hydroxyl radicals generated during the decomposition processes of ozone. This is typically true in water media, as in the case depicted in Ref. 29. However, here the test is carried out in vapor phase. Is the statement valid as well? Is there a relevant role of water in the system?
